# Melatonin Relations with Energy Metabolism as Possibly Involved in Fatal Mountain Road Traffic Accidents

**DOI:** 10.3390/ijms21062184

**Published:** 2020-03-22

**Authors:** Claus Behn, Nicole De Gregorio

**Affiliations:** 1Laboratory of Extreme Environments, Department of Physiology and Biophysics, Institute of Biomedical Sciences, Faculty of Medicine, University of Chile, Santiago 8380453, Chile; nicole.degregorio@gmail.com; 2Faculty of Medicine, Campus Los Leones, San Sebastián University, Providencia, Santiago 7500000, Chile

**Keywords:** melatonin, dysrhythmia, mountain road death

## Abstract

Previous results evidenced acute exposure to high altitude (HA) weakening the relation between daily melatonin cycle and the respiratory quotient. This review deals with the threat extreme environments pose on body time order, particularly concerning energy metabolism. Working at HA, at poles, or in space challenge our ancestral inborn body timing system. This conflict may also mark many aspects of our current lifestyle, involving shift work, rapid time zone crossing, and even prolonged office work in closed buildings. Misalignments between external and internal rhythms, in the short term, traduce into risk of mental and physical performance shortfalls, mood changes, quarrels, drug and alcohol abuse, failure to accomplish with the mission and, finally, high rates of fatal accidents. Relations of melatonin with energy metabolism being altered under a condition of hypoxia focused our attention on interactions of the indoleamine with redox state, as well as, with autonomic regulations. Individual tolerance/susceptibility to such interactions may hint at adequately dealing with body timing disorders under extreme conditions.

## 1. Introduction

The biosphere oscillates with a near 24-h precision, according to light (and temperature) changes determined by Earth rotation [1,2,3]. Along eons, living beings learned to anticipate recurrent environmental changes and to organize their physiology and behavior correspondingly [4,5,6,7]. A clock network of transcription-translation feedback loops allows mutually antagonistic processes to be temporally segregated. Gluconeogenesis and glycolysis, thus, can respectively prevail during resting and active phases of the day [8]. Individual chronotypes [9] match with the period length of gene expression rhythms in their fibroblasts maintained in culture out of the body [10]. Such rhythms relate to exercise performance being achieved at different daytime [11]. Primordial timekeeping of creatures contrasts, however, with widespread habits of current “24-h society” [2]. People try to keep moving, more or less spinelessly following the monotonous mandate of machines. Perturbations of primordial timekeeping however, associate with mental and physical shortfalls, disease and even shortening of life expectancy [12,13,14]. Metabolic dysrhythmia increases the risk of cardiometabolic diseases [15]. Even perturbations in the timing of food intake may affect metabolic homeostasis, potentially resulting in circadian dysrhythmia between organs [16]. A high-fat meal consumed at the end of the active period of the day, leads to increased adiposity, decreased glucose tolerance, hyperinsulinemia, and decreased cardiac function, as compared to mice fed the same high-fat meal, but during the beginning of the active period [17]. Time-restricted feeding, on the other hand, shifts the peak of respiratory quotient oscillations and increases their amplitude as compared to ad libitum-fed mice [18], and tends to prevent metabolic disease [19]. Circadian dysrhythmia implicates decrements of performance [20,21,22,23] that, in turn, represent a main cause of accidents. The latter becomes especially relevant in the case of human work realized in unusual environments. Large scale mining in Chile, mostly realized above 3000 m about sea level (m asl), thus, reached in 2013, an annual incidence of 25 fatal accidents/100,000 workers [24]. This number quintuplicates the mean incidence of deaths at work in Chile, the latter number already doubling the corresponding average in OECD countries. Present day work-related time schedules often collide also in habitual environments with endogenous rhythmicity of body functions. We have still much to learn concerning how to balance natural body timekeeping with modern lifestyle and ambitions. An increased interest in rhythms is, therefore, justly noticed in health care and medicine [25]. 

## 2. Rhythm Generation

Life implicates and depends on rhythms. Energy flowing through living beings generates linear and/or cyclic deviations from thermodynamic equilibrium [26] “Nondecaying, repetitive phenomena”, or cycles, determine the chemical organization of living matter [27]. Cycles also occur in simple organic chemical processes like the Belousov-Zhabotinskii reaction [28]. Oscillatory reactions often involve positive free enthalpy (Durup, 1979 cited by [29]). An oscillating signal requiring less energy implicates a gain in precision of control [30]. Biological control based on periodical events, thus, becomes energetically more advantageous and more efficient than that relying on steady-state reactions [30]. Phased with the environment at more or less constant period length, endogenous oscillators influence the genome [5] and by means of the latter, also affect functions like cell growth, protein synthesis, stress responses, and intermediary metabolism [31]. Timekeeping of energy metabolism and related body functions is mostly under the control of transcriptionally based, cell autonomous mechanisms in step with post-translational processes. Interlocked by transcriptional feedback loops, gene products like CLOCK and BMAL1 drive biological clocks positively, while others like CRY1-2, PER1-3 and REV-ERBα do it negatively [5,7,31,32]. Individual phase differences in PER3 expression, thus, correlate with timing of sleep during a constant routine [33]. The mechanisms by which the multiple feedback loops are integrated remain largely to be specified. Uncertain are also the post-translational mechanisms possibly involved in clock regulation [14,34,35,36]. Clock components (such as BMAL1 and CLOCK) may, moreover, additionally affect gene expression and metabolic processes, apparently not related to their proper timing function [15,37]. The mechanism by which clock protein stability connects with circadian period length is not yet exactly known [38]. Clinical consequences of period shortening in circadian cycles are evidenced in sleeping sickness caused by *Trypanosoma brucei* [39]. Noticeably, however, cycling also occurs in the absence of a genome, as shown by intracellular K^+^ levels in human red cells [40]. However rhythms are caused, they allow the necessary time keeping in living beings, or as having stated by Joseph Bass (2017): *As energy metabolism being in constant flux, there would be time in biochemical processes, as there is in a central train station* [41].

## 3. Rhythms, an Information System

Periodic processes tend to function in coupled entities [42,43,44]. The phase response of a single oscillator, thus, can sustain timing behavior at the multi-oscillator level [45]. From daily repetition of steady phase relationships among multiple clocks, a temporal order of the whole organism can arise. Interrelated clocks seem to constitute together a complex information handling system, keeping the organism as a whole in step with the challenges imposed by a periodically changing environment [30,46,47,48,49]. Thus, also the central circadian system represents an ensemble of oscillators crossing different brain regions [31]. Rhythms of body temperature, thus, also relate to locomotor activity [50,51], both together determining exercise performance [52,53]. Activity of brown adipose tissue seems, moreover, be involved in this context [54,55,56]. Oscillations of body core temperature, noticeably, coinciding with the daily sleep-wake cycle, make a phase advance already in human midlife [46,57,58]. Daily repeating phenomena, thus, coordinate widespread functions like temperature control [59,60,61], cardiovascular performance [62], autonomic [63], and endocrine regulations [49,64,65], as well as, behavior [52,53,66,67]. Biological oscillators, thus, effectively contribute to sustain homeostasis [30,68] and related functions like preconditioning (hormesis) and learning [69,70,71,72,73,74]. Even everyday coupling of rhythms between subjects has a neurophysiological substrate [75]. 

## 4. In the Beginning There Was Light

Environmental rhythms are coupled in homeothermic vertebrates with endogenous oscillators by a light sensitive neurohumoral network that includes retina, retinohypothalamic tract, nucleus suprachiasmaticus, pars reticularis, and the epiphyseal hormone melatonin [76,77,78,79,80,81,82]. Melatonin, an ubiquitous indoleamine, crucially influences the life of bacteria [83,84], unicellular algae [85], higher plants [86,87,88,89], and animals. The synchronizing role of melatonin in higher animals extends from the central nervous system up to peripheral oscillators in the cardiovascular system, skin, liver, adrenals, and various primate fetal tissues. Beside its crucial role as a circadian rhythm regulator, melatonin functions as a free radical scavenger and antioxidant agent [84], as well as, a neuroprotective [90], anti-inflammatory [91,92], and immunoregulating molecule [91], and even an oncostatic factor [84,93,94]. Melatonin, moreover, enhances carotid body chemoreceptor sensitivity [95] and lowers body temperature [77,96,97]. A normal daily schedule, may strengthen, on the other hand, circadian rhythmicity even under conditions of compromised light/dark cycles [98]. Melatonin not only responds to circadian light/dark cycles and seasonal differences of light’s impact on plants and animals [14]. The indoleamine appear also be regulated by food intake [31], involving factors like ghrelin [99] and orexin A [100]. Additional cues for clock functions include learned patterns like physical exercise routines [101]. Noticeably, cycles in tissue oxygenation appear to synchronize cellular clocks depending on hypoxia-induced factor 1-alpha (HIF-1α) [102]. 

Changes in environmental light [103], particularly concerning day length—as well as light intensity [104] and wavelength prevalence [105]—affect circadian rhythmicity. Light/dark changes affect almost all body functions [80,106,107,108]. Light in the night disrupts endocrine rhythms of living beings accustomed to being active during the day [109,110]. Light emerging from human affairs distorts bird reproductive behavior and mating patterns [111]. Melanopsin, an ancient bistable photopigment contained in “intrinsically photosensitive retinal ganglion cells” (RGCs), initiates the signal transduction by which light drives circadian rhythms [112]. Peak activation of intrinsically photosensitive RGCs occurs by light in the range of 460–480 nm, coinciding with the maximal absorption spectrum of melanopsin. Blue-light (480 +/− 20 nm) triggers the melanopsin photoreceptor system related to circadian rhythm control [113]. RGCs convey light induced signals along the retinohypothalamic tract to a master clock in the suprachiasmatic nuclei (SCN) of the basal hypothalamus for ulterior control of peripheral tissues. The master clock in SCN sensitizes peripheral molecular clocks by neurohumoral means to the light/dark cycle, thereby integrating circadian rhythmicity at the whole-body level [112]. The principal photic resetting cue to the SCN is in the 470 to 480 nm range, the range of wavelengths responsible for maximal RGC activity. Prior exposure to long-wavelength light may enhance SCN responses to 480 nm light [114]. 

However, it is in the epiphysis, or pineal gland where—according to Descartes—the soul interacts with the body via “vital spirits” [115,116]. The pineal gland provides melatonin (N-acetyl-5-methoxytryptamine) by N-acetyltransferase mediated cleavage of serotonin, the latter derived from the plant amino acid tryptophan [117]. Coincident with the absorption spectrum of melanopsin, blue light suppresses nocturnal melatonin secretion with maximal efficiency in the 446 to 477 nm range [118]. Filtration of a 10-nm bandwidth of light between 470 and 480 nm prevents light-mediated melatonin suppression. Blocking all wavelengths less than 530 nm, on the other hand, enhances melatonin secretion [119,120]. Melatonin cycles are altered, sleep is disrupted, and symptom prevalence is elevated in night workers [121]. Risks for two or more symptoms were 3.5 to 8 times greater among workers with sleep:work ratios < or =1 than those with ratios >1 [21]. Controlled reduction of short wavelengths in polychromatic light may prevent negative impacts on cardiac physiology without affecting cognitive performance and alertness in night-shift workers [122]. It must be considered, however, that the photic stimuli applied in this work included beside the short wavelength restriction, a very high irradiance level. This circumstance may relativize the above effect of short wavelength restriction. Interventions on melatonin secretion are difficult to apply in rotating shifts [123] and may have adverse effects on health [124].

A high altitude (HA)-related defect in blue axis vision and discrimination may affect non-visual brain responses to light, including defects in circadian rhythm control. Amplitude of circadian melatonin rhythm diminished and its relationship with respiratory quotient (RQ) weakened at a moderate altitude (Figure 1 of [125], with permission to be asked). Vision defects may be implied in melatonin rhythm alteration at HA. Exposure to HA alters color vision [126,127]. While protan (red) and deutan (green) axis discrimination seems still to be normal at 5400 m, tritan (blue) axis vision and discrimination are, on the contrary, reduced at HA [127,128,129]. Intense physical exercise seems to have a similar effect [130]. Melatonin secretion [131] and urinary 6-hydroxy-melatonin sulfate excretion [132] (derived from data contained in [125]), correspondingly, increase on acute exposure to HA. Summing up midday and midnight values of salivary concentration of melatonin renders a higher figure at 3700 m ASL than at sea level (Figure 2). The mechanism by which HA affects blue-light vision and, thereby, potentially also circadian rhythm control, remains to be elucidated. RGC have, indeed, been shown to be extremely hypoxia-sensitive [133]. Beta-adrenoceptors appear to trigger melatonin synthesis [134]. Hypoxia-related sympathetic stimulation may thus be also involved in diurnal increases of melatonin at high altitude. Overabundance of short-wavelength-enhanced light at HA [135] caused by Raleigh scattering [136] may be expected to induce protective mechanisms. Short-term neuronal plasticity in the retinohypothalamic tract synapses of suprachiasmatic nuclei [137], thus, may lead to synaptic depression [138]. Protection of SNC neurons by synaptic depression has also been advocated to favor clock adjustment in response to slow changes in the circadian light-dark cycle. Lack of light-induced melatonin suppression may be expected to cause somnolence at work and insomnia during the rest period, not only at HA, but in all working places without appropriate illumination. It should be noticed, however, that complex conditions are present in night-shift work, the real-life situation involving not only lighting. Light-induced melatonin suppression, however, clearly being impaired at HA, adds to an impressively large list of factors potentially inducing dysrhythmia at HA, including hypoxia, synaptic depression, sleep/wake cycle alterations, changes in food intake schedule, and last, but in our time, not least, the inescapable shift work. 

## 5. The Redox System, an Axis

Reactive oxygen species (ROS), lipid peroxidation, and nuclear factor kappa B (NF-κB) protein expression levels—as well as transvascular leakage—increase in the brains of rats exposed to a simulated altitude of 25,000 ft [139]. Mainly in response to stress, and at first glance, unrelated to circadian oscillation, melatonin is synthesized in the mitochondrial matrix of mice brains [140]. Once released into cytoplasm, melatonin activates a mitochondrial MT1 signal-transduction pathway. The MT1 pathway inhibits stress-mediated cytochrome C release and caspase activation, thereby protecting against neurodegeneration [90]. Melatonin, moreover, promotes the expression of sirtuin I (a histone deacetylase), thereby enhancing the amplitude of circadian oscillations and promoting survival [141]. Clock mutation in mice diminishes circadian pacemaker amplitude but leads resetting stimuli to be more efficient [142]. Receptors of melatonin in the SCN appear to require G-protein-coupled [90], inwardly rectifying potassium (GIRK) channels, thus participating in a widely distributed physiological neural communication system [143]. The potential usefulness of melatonin receptor-agonists to address sleep problems [144] may be seen in this context (but see also [145]). 

Tissue melatonin content increases in response to stressful conditions in plants [146], as well as in animals [131,132]. Beyond its role in supporting circadian rhythmicity, melatonin protects via antioxidant [84,147,148,149], anti-inflammatory [91,92], and oncostatic effects [94]. Melatonin limits ROS formation [150] and reduces photosynthesis, a known source of ROS [85]. Melatonin counteracts oxidative [151] and nitrosative [152] stress. The indoleamine scavenges ROS and nitrogen (RNS) reactive species [153]. Enhanced photo-consumption of melatonin by free radicals contributes to decrease the indoleamine on exposure to daylight in *Symbiodinium*, a dinoflagellate, but may have varied before rising irreversibly some 2.4 billion years ago during the Great Oxidation Event (GOE), by enhanced free radical production in relation with daily light/dark changes [85]. Melatonin also enhances antioxidant enzyme activities [154] and regenerates endogenous antioxidants like glutathione [155]. Relations of melatonin with oxygen may be traced back to emergence of the latter gas in the Earth’s atmosphere [83]. Atmospheric oxygen raised irreversibly during the GOE some 2.4 billion years ago [156]. Circadian melatonin oscillations in *Symbiodinium* are thought, indeed, not to be caused by endogenous circadian control, but rather by variations in photo-consumption of atmospheric oxygen [85]. 

Oxygen shortage occurring at HA may serve as a model for effects of cardiovascular and respiratory failures, strenuous physical exercise, pregnancy, ageing, inflammation, and terminal cancer. Hypobaric hypoxia of HA implicates alterations of energy metabolism [157], including oxidative stress [158]. ROS mediate, at least in part, tissue damage related to hypoxia and subsequent reoxygenation [159]. Repair of hypoxia related tissue damage requires energy. Lack of oxygen decreases the production of ATP but, for repair, concomitantly also increases energy demand. Reducing energy requirements in the presence of oxygen shortage, thus, can prevent hypoxia from occurring. The potential effects of hypoxia on mental and physical work capacity thus may be mitigated by increasing the availability of antioxidants. Lack of oxygen is also known to blunt the amplitude of circadian oscillations of oxygen consumption [160,161], with potential consequences for almost all body functions [60,161]. In adult rats, hypoxia (10.5% O_2_ for three days) reduced the amplitude of daily basal temperature oscillations by 55% in adult female rats, and 22% in adult male rats [161]. Hypoxia-related redox alterations may be suspected to be involved in HA-related dysrhythmia. H_2_O_2_ cycles have recently been found to be essential for clock function, as related to energy metabolism [162] see also [163,164,165,166,167]. Shift work, disordered food intake, and melatonin-related neurohumoral dysrhythmia may conflagrate in decrements of mental and physical performance occurring on exposure to HA. Normalizing redox cycles at HA may improve dysrhythmia under the latter condition.

Hypoxia, the lack of oxygen, as related to ATP requirements [168] affects biological clocks [169,170,171,172]. Hypoxia alters circadian rhythms of *Drosophila* [173], rats [174,175], and humans [176,177]. Circadian patterns of gene expression [178] and mitotic activity [179] are also affected by hypoxia. Adult rats having been exposed to hypoxia during their gestation, show diminished activity levels, phase-advanced activity rhythms, and delayed adjustment to light–dark perturbation [180]. Relative amplitude of daily oscillations is nearly constant among species [181,182], but appears to be drastically decreased by hypoxia. Hypoxia affects circadian oscillations of body temperature and metabolic rate [171]. As the latter variables influence almost all body functions [60], hypoxia may perturb all temperature dependent functions [172]. Acting as a hypothermic agent [77,96,97], melatonin may also protect.

Generating a surplus of electrons, hypoxia promotes free radical reactions [183]. ROS affect the structure of lipids [159,184], proteins [185,186], and nucleic acids [187]. Malondialdehyde (MDA), a lipid peroxidation product, relates in exhaled breath condensate (EBC) with severity of acute mountain sickness (AMS) in climbers exposed to an altitude of 5000 m asl [184]. MDA concentration in EBC also rises in response to acute cycloergometric exercise realized at 2200 m asl, but not at 670 m asl [184]. Lipid peroxidation correspondingly increases in response to physical exercise performed while the inspired fraction of oxygen is lowered to 0.16 [188]. Hypoxia also increases eicosanoid plasma concentration [189]. Extending to microsomal membranes, lipid peroxidation may further enhance oxidative stress. Melatonin appears to modulate redox status in pulmonary vessels along gestation [190]. 

Hypoxia leads to a somewhat stereotypic sequence of events in rather enclosed organs like brain, lungs, and perhaps also testis. Lack of oxygen primarily affects active transport implicating cellular salt and water accumulation. Brain white matter volume correspondingly increases in response to a subject’s exposure for 22 h to HA [191] but see also [192]. An increase of cell volume increases tissue pressure in organs unable to expand. The increase of tissue pressure reduces vascular transmural pressure. The capillary bed thus diminished accentuates the lack of oxygen. Ensuing cell damage releases alarmins, activating NF-κB, an ubiquitous, pleiotropic, and pro-inflammatory transcription factor. NF-κB is activated by hypoxia [193,194,195], even in Hela cell cultures [196,197] and in vessels [198]. NF-κB activation by hypoxia also leads to activation of HIF-1α. Concomitant activation of HIF-1α triggers the expression of hundreds of genes. Noticeably, melatonin inhibits HIF-1α [151]. Melatonin protects gastroduodenal mucosae [199]. HIF-1α-mediated lesions of the gastrointestinal tract have been observed on acute exposure to HA [200]. By inhibiting the angiogenic effect of HIF-1α [201], melatonin may also be included in the arsenal of antineoplastic agents. Varicocele (VC), a dilatation of scrotal portion of pampiniform plexus and the internal spermatic venous system [202], occurs among Chilean miners with an incidence directly related to geographical altitude of their usual working site. In a cohort of 465 miners working at different levels A (<2400 m; *n* =167), B (3000–3900 m; *n* = 86) and C (>3900 m; *n* = 243) the incidence of Grade 1 VC (not visible, but palpable), was respectively 4.4%, 9.5% and 36.9% (Marchetti, N (Faculty of Medicine, University of Chile, Santiago, Chile). Personal communication. 2020). Experimental VC upregulates HIF-1α/BNIP3/Beclin1 autophagy-signaling pathway in testicular tissue, revealing a condition potentially promoting male infertility [203]. Repair of experimental VC, on the other hand, reverts the presence of HIF-1α in rat testis [204].

## 6. No Swing, No Pleasure, No Health

Humans. chronically ignoring their intrinsic rhythmicity. Feel sick [12], report fatigue and somnolence [205], suffer from mood disorders [108,206,207,208], and tend towards drug abuse [209,210]. Psychiatric morbidity appears to be elevated in extreme human chronotypes, even under normal working conditions [211,212,213]. Shift work, thus, affects health [108,214,215], decreases mental and physical performance, and may even shorten life expectancy [62,216,217,218,219,220,221]. Hormonal imbalances associated with shift work include alterations of leptin and insulin levels [221,222,223]. Lack of leptin decreases fatty acid metabolism in nonadipose tissues, including the myocard [224]. Energy homeostasis, thus disrupted, leads to oxidative stress and telomere attrition [225], and derives into pathologies like metabolic syndrome [219], obesity [219,223,226], and type 2 diabetes mellitus [227,228]. Both long- (e.g., adiposity) and short-term (e.g., glucose/lipid tolerance) metabolic homeostasis are altered by circadian clock disruption [15]. Feeding during sleep phase increases adiposity in wild-type mice [17]. Long-term costs of night-shift work [121], moreover, involve risks of developing cardiovascular disorders [62] including arterial hypertension [221,229,230]. Accelerated aging [231], cancer promotion [232,233,234,235], breast [236,237] and colorectal cancer [238] profile in this respect. Mechanisms involved in circadian rhythm generation and control—as well as environmental factors potentially affecting them—thus must be considered for prevention and intervention of shift work related risks.

Feeding dysrhythmia, hyperphagia, obesity, and evidence of metabolic syndrome occur in homozygous clock-mutant mice [223]. However, longstanding misalignments between endogenous rhythms and exogenous cues [239,240] can affect health and survival of living beings, from plants [43,86,87,88,240] up to higher animals [231,241]. Uncoupling of endogenous biological rhythms from light changes determined by rotations of the earth can affect almost all body functions [80,106,107,108,242]. Heart rate and locomotor activity of rodents—usually resting during the day—diminish in response to light applied in the night [243]. Light in the night disrupts endocrine rhythms [109,110,222]. Altering bird reproductive behavior and mating patterns by light emerging from human affairs distorts previously reliable quality-indicator traits [111].

## 7. Death in the Mountain

Death menaces at HA, particularly at extreme altitudes. Shift work and related sleep deprivation [244,245,246], food intake disorders, permanent alertness, as well as hypoxia, combine to involve body timekeeping disorders that substantially contribute to endanger any human work on the mountain [247,248,249,250]. Moreover, healthcare facilities along mountain roads are often scarce, implicating additional difficulties for rescue teams to arrive in time. Mining and astronomy are mainly realized at HA (above 3000 m asl) in Andean countries like Chile. Dangers due to dysrhythmia in the mountain concern particularly road traffic. At least three conditions are to be considered in road traffic at HA: velocity of ascent, the altitude reached, and time spent at HA [249]. Difficulties are typically noticed by drivers at 2500 m asl [247], or 3000 m asl upward [250]. Physiological conditions for commuting at HA should differ substantially between ascent and descent. Among deaths registered above 8000 m asl on Mt Everest over an 86-year period, only 10% occurred during ascent. On the other hand, during descent from the summit, the death rate of climbers nearly sextuples that of Sherpas [251,252]. Information provided by Atacama Large Millimeter Array Observatory (ALMA) and National Geology and Mining Service of the Chilean Government, (SERNAGEOMIN) allowed us to derive the incidence of fatal road traffic accidents at HA sites in Chile. Retrospectively it was possible to distinguish whether each accident occurred on ascent or descent as seen in Figure 3 (De Gregorio, N (Faculty of Medicine, University of Chile, Santiago, Chile). Unpublished results. 2020). Noticeably, not only the incidence of fatal road traffic accidents (Figure 3), but also that of cardiac arrhythmias [248] are higher on descent than on ascent. The incidence of cardiac arrhythmias on descent is, moreover, higher in drivers younger than 40 years than in older ones [248]. Figure 4, correspondingly, shows an inverse relationship between fatal road traffic accidents and age of the driver above 2500 m asl (De Gregorio, N (Faculty of Medicine, University of Chile, Santiago, Chile). Unpublished results. 2020) Dysrhythmia of melatonin at HA [125] may be considered in this respect, particularly concerning some not yet fully-explored relations of indoleamine with autonomic nervous system (ANS). Risk-determining hemodynamic features along ascent and descent, respectively, depend on dominance of the sympathetic and the parasympathetic branches of ANS. Hypoxia, the prevalent challenge on ascent, implicates sympathetic stimulation [253], the latter presumably associated with enhanced melatonin secretion [134]. An overabundance of melatonin—provoked by sympathetic dominance during ascent—will most probably impact potentiating vagal activity; the latter being enhanced by subsequent reoxygenation during descent. Oxygen availability, being restored along descent, leads, indeed, to a rebound of parasympathetic tonus [254]. Vagotonusinversely relates to age [255,256] and tends to be augmented by melatonin [257,258,259]. Enhanced vagal activity can lead to accidents by provoking failures in consciousness. Dangerous absences while driving may particularly be expected to occur in relation with combined vagal effects on heart rhythm [248], vascular tone [260], and blood glucose levels [261]. Bradycardia induced by enhanced vagal activity implies augmented vulnerability to generation of extrasystoles emerging from ectopic foci. Melatonin further enhances the interval between heart beats, at least in supine humans [262]. Depending on its receptors (MT1 and MT2), melatonin exhibits concentration related cardiovascular effects [263,264,265] that, not being less important for circulatory and metabolic pathology [266] are beyond the scope of the present review. The incidence of arrhythmic events in truck drivers is higher during descent than ascent [248]. Any arrhythmic event—and the related cerebral hypoperfusion—may trigger absences leading to accidents. Moreover, vagal hyperactivity may involve venous dilatation, an effect also potentiated by melatonin, promoting blood pooling in the legs. A compensatory postural tachycardia is, on the contrary, reduced by melatonin [267]. Reflex sympathetic responses to orthostatic stress are diminished by melatonin [257]. Vagal tendency to promote orthostatic collapse, thus, possibly being potentiated by melatonin, may successively curtail venous return, diastolic filling of the heart, cardiac output and, hence, also cerebral perfusion. Vagal hyperactivity may also enhance insulin secretion. Cognitive shortfalls promoting accidents on descent may thus, also result from hypoglycemia. Triggering tidal secretion of melatonin, by transcutaneous vagus nerve stimulation has been shown to lower blood glucose levels in Zucker fatty rats [268].

## 8. Conclusions

All in all, defective melatonin interactions with energy metabolism, particularly affecting redox status and autonomic regulations, may substantially contribute to current death tolls for working in extreme environments (Figure 5). 

A more detailed knowledge on the mechanisms determining individual susceptibility under such conditions will give, in this respect, some outlook into a brighter future.

## Figures and Tables

**Figure 1 ijms-21-02184-f001:**
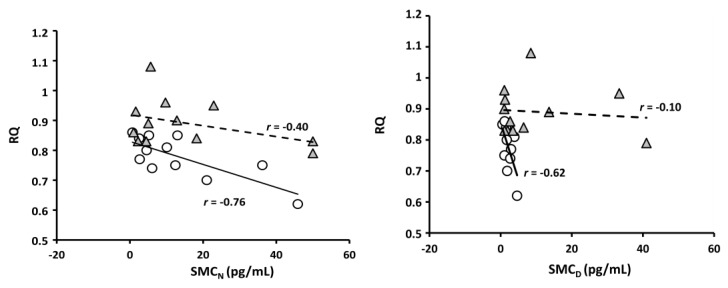
Salivary melatonin concentration (SMC) as related to respiratory quotient (RQ) (permission to be asked, Tapia et al., 2018) in *n* = 12. At the left, the relation between RQ and SMC at 12:00 PM (SMC_D_) at sea level (white circles) and at high altitude (HA) (grey triangles). On the right side, the same relation at 12:00 AM (SMC_N_), also at sea level and HA.

**Figure 2 ijms-21-02184-f002:**
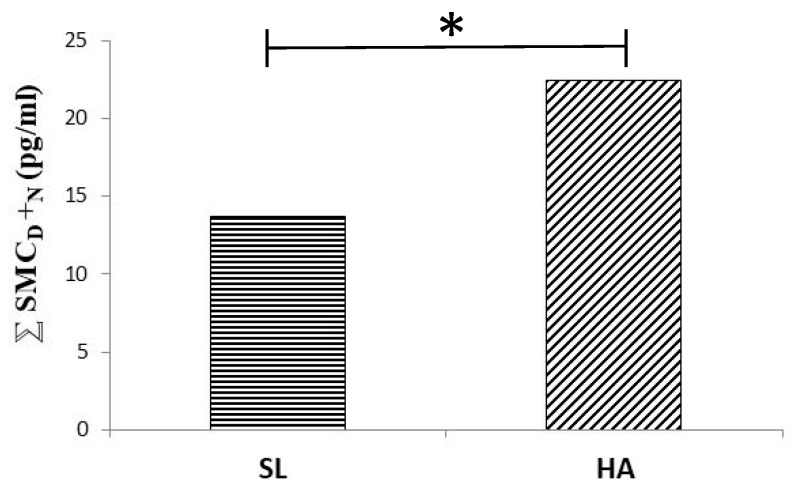
The sum of total (night and day) SMC at SL (left) and HA (right). Asterisk represents statistical difference (*p* < 0.05).

**Figure 3 ijms-21-02184-f003:**
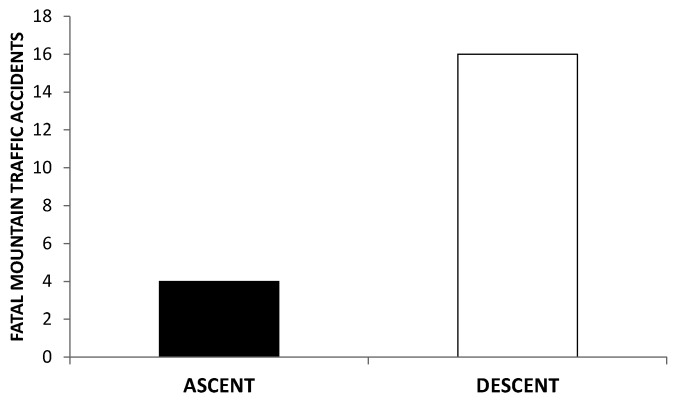
Fatal road traffic accidents in Chilean mining (*n* = 18) and Atacama Large Millimeter Array (ALMA) Observatory (*n* = 2) between the years 2005 and 2017. Column on the left (*n* = 4) and the right (*n* = 16), respectively, indicate the number of cases that occurred during ascent and descent (De Gregorio, N (Faculty of Medicine, University of Chile. Santiago, Chile). Unpublished results. 2020).

**Figure 4 ijms-21-02184-f004:**
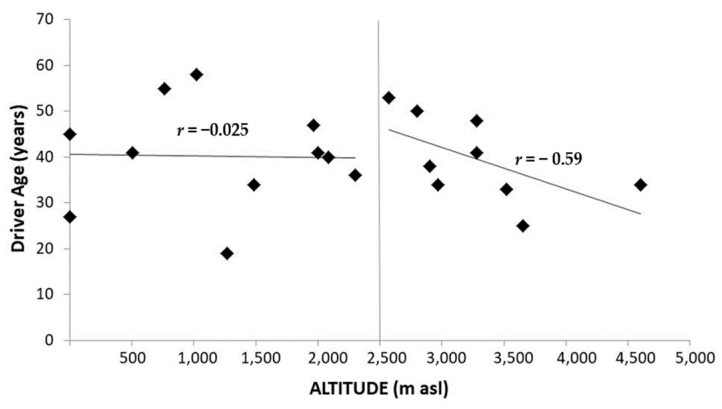
Fatal road traffic accidents in Chilean mining (*n* = 18) and ALMA Observatory (*n* = 2) between the years 2005 and 2017, comparing the age of the deceased driver in terms of the altitude at which the accident occurred. The younger the subjects are, they appear to be more susceptible to suffer fatal accidents at altitudes from 2500 m ACL upward (*r* = −0.59) (De Gregorio, N (Faculty of Medicine, University of Chile. Santiago, Chile). Unpublished results. 2020). Notice that subjects younger and older than 40 years appear rather evenly distributed on the right-hand side of the graph.

**Figure 5 ijms-21-02184-f005:**
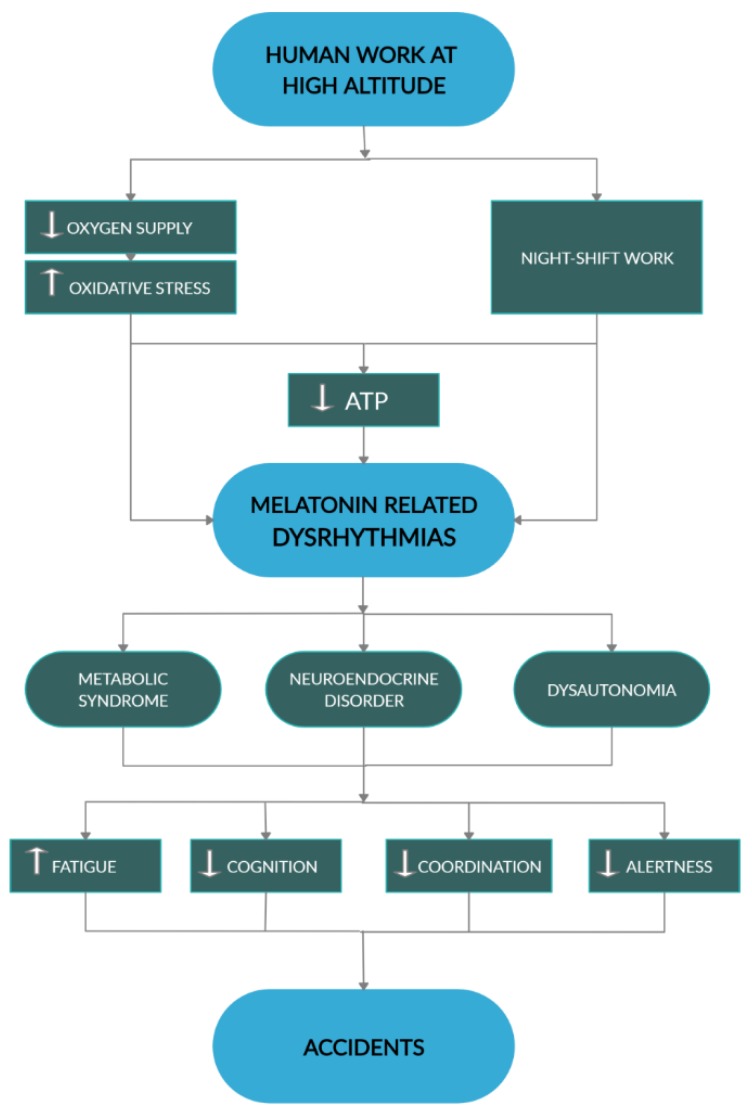
This diagram summarizes dysrhythmia centered mechanisms by which human work at HA may promote accidents. White arrows, respectively indicate increase and decrease. Black arrowheads indicate direction of processes.

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
