# Peer review of "Melatonin Relations with Energy Metabolism as Possibly Involved in Fatal Mountain Road Traffic Accidents"

_ijms, 2020, doi:10.3390/ijms21062184_

Round 1
Reviewer 1 Report
The authors discuss an interesting proposition relating working in high altitude and specifically road accidents to melatonin and biological rhythms disruption in this extreme condition. The discussion involves a good review about biological rhythms origin and mechanisms involved in its regulation, hypoxia effects on human beings and cellular metabolism. Along the text the authors argue in favor of a biological rhythms disfunction related to high altitude and construct successfully the link with working and road accidents.
Considering the complexity of the subject, more illustrative figures would be very helpful for the readers understanding and a global view of the mechanisms involved.
Comment 1:
I strongly recommend to insert a figure with the physiological mechanisms involved in high altitude symptoms and the pathways where rhythmic disruption and melatonin could be involved. Otherwise, figures 2 and 3 do not add more information than already presented in the text.
Comment 2:
Page 4 (line 151) to page 5:
The Tritan axis impairment is suggested to be related to blue cone (involved in blue color vision and discrimination) susceptibility to hypoxia, but the projections to the timekeeping system arise in a deeper layer, in intrinsically photosensitive retinal ganglion cells (RGC). Otherwise, other mechanisms, as increased sympathic ativation under hypoxia could be involved in increased diurnal melatonin in high altitude.
I suggest to discuss references, if any, about RGC compromising under hypoxia.
Comment 3:
Page 4, line 147: “Controlled reduction of short wavelengths in polychromatic light may prevent negative impacts on cardiac physiology without affecting cognitive performance and alertness in night-shift workers [122].”
Page 5, line 168: “Lack of light-induced melatonin suppression can be expected to cause somnolence at work, and insomnia”.
Night-shift work is a complex situation that has its deleterious effects related to both circadian misalignment and sleep deprivation. Reduction of short wavelenghts (please, add the information that this refers to blue spectra) is expected to decrease the photoinhibition of melatonin production, which didn´t show impact on alertness and cognitive performance, contradicting the hypothesis presented in line 168, page 5. However, the experimental condition cannot reproduce all the real life factors involved in night-shift work. This should be discussed in a broader approach.
Comment 4:
About Page 9-10:
MT1 and MT2 receptors are found in cardiovascular organs including peripheral and central blood vessels, heart, kidneys, adrenal glands. Stimulation of vascular melatonin receptors induces vasoconstriction (MT1) or vasodilation (MT2) depending on its concentration. Melatonin may modulate baroreceptor reflex control of heart rate via melatonin receptors in the area postrema. Hence melatonin action on cardiovascular parameters can involve direct action mechanisms besides the vagal activity modulation. I suggest to include these points in the discussion. Some references are below.
1-Doolen, S., Krause, D.N., Dubocovich, M.L., Duckles, S.P. Melatonin mediates two distinct responses in vascular smooth muscle. Eur. J. Pharmacol., 1998, 345: 67–69.
2-Masana, M.I. MT2 Melatonin Receptors Are Present and Functional in Rat Caudal Artery. J. Pharmacol. Exp. Ther., 2003, 302: 1295–1302.
30Campos, L.A., Pereira, V.L., Muralikrishna, A., Albarwani, S., Brás, S., Gouveia, S. Mathematical biomarkers for the autonomic regulation of cardiovascular system. Front. Physiol., 2013c, 4 OCT: 1–10
Comment 5:
Figure 4: the statement in the text and figure 4 that younger subjects would be more susceptible to fatal accidents in high altitude depend on the information about the amount of drivers in each age in the referred data bank. The results presented could be explained by a greater amount of younger drivers in HA. Please include the information about drivers age distribution.
Minor issues
Page 2, line 32: “Gluconeogenesis and glycolysis, thus, can respectively prevail during resting and active periods of the day” – resting and active phases, not period.
Page 4: Include the meaning of the initials HIF-1α in line 120 (hypoxia--induced factor 1-alpha)
Author Response
Dear Reviewer 1,
Thank you very much for your exhaustive revisión and most valuable suggestions. Please find included below our answers to your comments.
Comment 1:
I strongly recommend to insert a figure with the physiological mechanisms involved in high altitude symptoms and the pathways where rhythmic disruption and melatonin could be involved. Otherwise, figures 2 and 3 do not add more information than already presented in the text.
Answer to Comment 1:
According to your suggestion, we included a flow diagram (Fig. 5) indicating possible dysrhythmia centered pathways by which human work at high altitude may promote accidents.
Comment 2:
Page 4 (line 151) to page 5:
The Tritan axis impairment is suggested to be related to blue cone (involved in blue color vision and discrimination) susceptibility to hypoxia, but the projections to the timekeeping system arise in a deeper layer, in intrinsically photosensitive retinal ganglion cells (RGC). Otherwise, other mechanisms, as increased sympathic ativation under hypoxia could be involved in increased diurnal melatonin in high altitude.
I suggest to discuss references, if any, about RGC compromising under hypoxia.
Answer to Comment 2:
Intrinsically photosensitive retinal ganglion cells (RGC) have indeed been shown to be extremely hipoxia sensitive. An increase of melatonin availability can, accordingly, be expected as a consequence of hypoxia induced RGC damage, as well as, of hipoxia induced sympathetic activation (Page 5, Line 166-168).
Comment 3:
Page 4, line 147: “Controlled reduction of short wavelengths in polychromatic light may prevent negative impacts on cardiac physiology without affecting cognitive performance and alertness in night-shift workers [122].”
Page 5, line 168: “Lack of light-induced melatonin suppression can be expected to cause somnolence at work, and insomnia”.
Night-shift work is a complex situation that has its deleterious effects related to both circadian misalignment and sleep deprivation. Reduction of short wavelenghts (please, add the information that this refers to blue spectra) is expected to decrease the photoinhibition of melatonin production, which didn´t show impact on alertness and cognitive performance, contradicting the hypothesis presented in line 168, page 5. However, the experimental condition cannot reproduce all the real life factors involved in night-shift work. This should be discussed in a broader approach.
Answer to Comment 3:
The observation that short wavelenght restriction in polychromatic light may not affect cognitive perfomance and alertness in night-shift workers (122) may be somewhat biased the high irradiance levels, the authors applied in their experimental model. The latter may indeed not exactly reproduce “all the real life factors involved in night-shoft work”
Comment 4:
About Page 9-10:
MT1 and MT2 receptors are found in cardiovascular organs including peripheral and central blood vessels, heart, kidneys, adrenal glands. Stimulation of vascular melatonin receptors induces vasoconstriction (MT1) or vasodilation (MT2) depending on its concentration. Melatonin may modulate baroreceptor reflex control of heart rate via melatonin receptors in the area postrema. Hence melatonin action on cardiovascular parameters can involve direct action mechanisms besides the vagal activity modulation. I suggest to include these points in the discussion. Some references are below.
Answer to Comment 4:
According to your suggestion, we included the references you indicated. Adittionally, we mencioned a recent review referring to melatonin related modulation of cardiovascular and metabolic pathology (Page 10, Line 378-380). We apologyze for not entering in details of melatonin receptors physiology as this topic may exceed the scope of this review.
Comment 5:
Figure 4: the statement in the text and figure 4 that younger subjects would be more susceptible to fatal accidents in high altitude depend on the information about the amount of drivers in each age in the referred data bank. The results presented could be explained by a greater amount of younger drivers in HA. Please include the information about drivers age distribution.
Answer to Comment 5:
As can be observed, on the right hand side of Fig.4, the distribution of subjects older and younger than 40 years is nearly the same (4 vs. 5) at very high altitude (> 2,500 m asl).
Minor issues
Page 2, line 32: “Gluconeogenesis and glycolysis, thus, can respectively prevail during resting and active periods of the day” – resting and active phases, not period. OK, thank you.
Page 4: Include the meaning of the initials HIF-1α in line 120 (hypoxia--induced factor 1-alpha). OK, thank you.
Reviewer 2 Report
The manuscript entitled “Melatonin relations with energy metabolism as possibly involved in fatal mountain road traffic accidents” by Claus Behn and Nicole De Gregorio is a large-scale review article discussing the relationship between melatonin cycle and fatal mountain road traffic with elaborate logic, meticulous storytelling, and abundant literatures. Discussions about biological rhythm and redox system are not only scientifically accurate and sound, but also fascinating. Besides, these basic backgrounds are naturally connected to human behavior and pathology, which eventually leads to the understanding of the higher incidence of traffic accidents at high altitude.
A minor point: It seems that Fig. 4 suggests that the younger the drivers are, they are more susceptible to suffer fatal accidents at altitudes from 2,500 m asl upward. Is there any chance that there is a bias in driver age at higher altitude and only younger drivers go to higher place?
Author Response
Dear Reviewer 2,
thank you very much for your favorable comments referring to our manuscript. Please find included below our answer to your comment.
Comment 1
A minor point: It seems that Fig. 4 suggests that the younger the drivers are, they are more susceptible to suffer fatal accidents at altitudes from 2,500 m asl upward. Is there any chance that there is a bias in driver age at higher altitude and only younger drivers go to higher place?
Answer to Comment:
In the legend of Fig. 4 we have included a sentence noticing that subjects older and younger than 40 years are nearly evenly distributed (n = 4 vs. 5) at altitudes above 2,500 m asl (right hand side of the graph).